# An Assessment of Dietary Intake, Feeding Practices, Growth, and Swallowing Function in Young Children with Late-Onset Pompe Disease: A Framework for Developing Nutrition Guidelines

**DOI:** 10.3390/nu17111909

**Published:** 2025-06-01

**Authors:** Surekha Pendyal, Rebecca L. Koch, Harrison N. Jones, Priya S. Kishnani

**Affiliations:** 1Division of Medical Genetics, Department of Pediatrics, Duke University Medical Center, Durham, NC 27710, USA; rebecca.koch@duke.edu (R.L.K.); priya.kishnani@duke.edu (P.S.K.); 2Head and Neck Surgery & Communication Sciences, Duke University School of Medicine, Durham, NC 27710, USA; harrison.jones@duke.edu

**Keywords:** late-onset Pompe disease, swallowing function, growth, dietary intake, nutrition

## Abstract

Newborn screening (NBS) is leading to the diagnosis of a large number of children with late-onset Pompe disease (LOPD), yet many remain asymptomatic until later years. A high-protein, low-carbohydrate diet is recommended for adults with LOPD. Nutrition guidelines are not available for young children. **Methods**: 37 children with LOPD aged 1–6 years participated. Early diet history, feeding practices, and 24 h dietary intake were collected via questionnaire. Anthropometric measurements, blood creatine kinase (CK), blood urea nitrogen (BUN)/creatinine ratio, and urine glucose tetrasaccharide (Glc4) were collected at clinic visits. A subset of 19 children received a clinical feeding assessment (CFA). **Results**: All patients derived their nutrition orally. Breastfeeding was successfully initiated in 73% of infants. Body weight ranged between 3 and 99% and height ranged from 4 to 97%. A tendency to be overweight and obese was noted in older children with LOPD. A total of 24% of the children who had CFA were diagnosed with dysphagia that was typically mild in severity and rarely affected their ability to eat a normal diet. Limiting added sugar and processed foods was the most widely used dietary practice followed by encouraging protein. Protein intake was three–four times higher than the recommended dietary intake (RDA). A high BUN/creatinine ratio was observed in some children, which may indicate incompatibility with protein intake and need for individualizing the diet. **Conclusions**: The results of this study provide a framework for developing future nutrition guidelines for children with LOPD by performing an individualized assessment of dietary intake, growth, feeding/swallowing, and laboratory parameters.

## 1. Introduction

Pompe disease is a neuromuscular disorder caused by a deficiency of the enzyme acid alpha-glucosidase (GAA)—an enzyme responsible for hydrolyzing lysosomal glycogen to glucose. Pathological glycogen storage is the hallmark of the disease and disrupts the metabolism and function of various cell types, especially muscle cells, leading to cardiac, motor, and respiratory dysfunction. The spectrum of Pompe disease spans two main forms: classical infantile-onset (IOPD) and late-onset (LOPD). IOPD, caused by complete or almost complete GAA deficiency, presents at birth and leads to premature death by the age of 2 years without treatment. LOPD results from residual GAA enzyme deficiency and appears in infancy, childhood, adolescence, or adulthood with a wide spectrum of involvement including muscle weakness and respiratory problems [1]. In 2015, Pompe disease was added to the recommended uniform screening panel (RUSP) for newborn screening (NBS) in the USA. Forty-six states in the USA now include Pompe disease in their NBS panel. Results from these programs show the combined incidence rates of IOPD and LOPD to be ~1/20,000, which is significantly higher than the traditional estimate of 1/40,000 births [2,3].

Enzyme replacement therapy (ERT) with alglucosidase alfa has been the mainstay of treatment for Pompe disease for both IOPD and symptomatic LOPD patients since its approval by the FDA in 2006. Two next gene therapies avalglucosidase alfa and cipaglucosidase alfa with miglustat are now also approved for Pompe disease, and gene therapy trials are underway. While ERT has shown to have a remarkable effect on reversing cardiac hypertrophy and improving survival, it is not fully able to prevent neuromuscular weakness and progressive deterioration in survivors [4]. Due to the limitations of ERT, the development of next-generation therapies and implementation of ancillary therapeutic approaches are important in optimizing the care of Pompe patients [5]. A high-protein, low-carbohydrate diet with exercise therapy continues to be recommended as adjunct therapy in adults with LOPD [6], with the intention of providing less substrate to form glycogen and increasing utilization of amino acids for preserving muscle function. However, the impact of this diet on young children with LOPD is not known. Moreover, with an increase in number of patients with LOPD being diagnosed by NBS, it is not clear what dietary regimens, if any, these children are using. The purpose of the present study was to assess the feeding practices, dietary intake, feeding/swallowing, and growth in young children with LOPD diagnosed via NBS, and provide a framework for the development of nutrition guidelines in this population.

## 2. Materials and Methods

Children who screened positive for Pompe disease via NBS and were confirmed to have the late-onset form based on the absence of cardiac involvement, presence of residual enzyme activity, and with GAA variants consistent with a LOPD phenotype (such as c.-32-13T > G, c.2188G > T, c.1935C > A, c.1726G > A, or c.118C > T in homozygosity or compound heterozygosity) via genetic testing were included in the study after obtaining written informed consent as part of two IRB-approved studies on Pompe disease: Pro00010830 and Pro00100223. A questionnaire (Appendix A) was developed to collect information on early infant nutrition and feeding practices, nutrition history in terms of foods encouraged and discouraged once the child had started eating solids, and present dietary intake. Parents completed the questionnaire along with a 24 h diet recall at clinic visits or via email. A hand portion guide [7] with picture diagrams was provided to parents to gauge portion size of foods consumed by patients for completing the diet recall. Diet information was collected and analyzed by a metabolic dietitian with expertise in glycogen storage diseases using MetabolicPro (Genetic Metabolic Dietitians International, Hillsborough, NC, USA).

Patients were referred to Duke Speech Pathology for a clinical feeding assessment (CFA) at their first visit. The CFA included feeding/swallowing history, oral motor examination, feeding trials with a variety of age-appropriate items, determination of the Functional Oral Intake Scale—Pediatric (FOIS-P) score [7,8], and administration of age-appropriate proxy-reported outcome measures. Proxy measures included the Neonatal Eating Assessment Tool—Bottle Feeding (NeoEAT Bottle Feeding) [9,10,11] for bottle-feeding infants less than 7 months of age, the Neonatal Eating Assessment Tool—Breastfeeding (NeoEAT Breastfeeding) for breastfeeding infants less than 7 months [9,12,13], and the Pediatric Eating Assessment Tool (PediEAT) for older children [14,15,16]. When appropriate, a videofluoroscopic swallowing study (VFSS) was recommended.

Growth parameters and clinical chemistry labs were collected at clinic visits as part of routine clinical care. Growth percentiles were standardized according to Center for Disease Control (CDC) growth curves. Dietary intakes were compared to Dietary Reference Intake (DRI) guidelines issued by the Food and Nutrition Board of the National Academies of Sciences Engineering and Medicine, including the recommended dietary allowance (RDA) and acceptable macronutrient distribution range (AMDR). Collected data were grouped by age of children (Table 1): 1 to <2 (*n* = 11), 2 to <4 (*n* = 14), and 4 to <6 years (*n* = 12).

Blood urea nitrogen (BUN), creatinine, BUN: creatinine ratio, creatine kinase (CK), and urine Glc4 were collected either at the clinic visit or within 4 weeks of when the nutrition data were collected at the medical center where the patient received care. Because the normal ranges of some laboratory tests varied between centers, data were interpreted as within normal range or out of range to their respective normative data.

Figure 1A–F illustrates the growth, distribution of dietary intake, and the laboratory data at different ages.

## 3. Results

Thirty-seven children (19 M and 18 F) diagnosed with LOPD via NBS in 19 different states were referred to Duke University Metabolic Genetics clinic for follow-up care and their parents provided consent for study participation. Consistent with previous reports on the US population [3,17,18,19,20], the IVS-c.-32-13T > G splice site variant was the most commonly observed variant in this group with 28/37 (76%) of children having this variant in compound heterozygosity and 5/37 (13.5%) children being homozygous for this variant. The remaining four children (11%) in the study did not carry this variant. The age range of patients was 1 year to 5.6 years (median age: 3.5 years). At the time of the study, 11/37 (30%) patients were on ERT, which was initiated between 3 weeks to 4.3 years (median age: 11 months). Of the thirty-seven children, thirty-four were born at term and three were born between 31 and 36 weeks. One child (born at 31 weeks) was the product of a twin pregnancy. Initiation of ERT was based on the clinical judgement of the medical providers.

### 3.1. Early Feeding History

Breastfeeding was initiated in 27/37 infants (73%) and 10/37 (27%) were given infant formula at birth. Some of the breastfed infants (6/27, 22%) continued to be exclusively breastfed until weaning at ~1 year while the majority (21/27, 78%) received infant formula in addition to breast milk at some point before weaning. Standard infant formulas as well as hypoallergenic US-manufactured infant formulas which contained lactose, maltodextrin, corn syrup solids, and a few with sucrose, were most commonly used. A few children (4/31, 13%) consumed European-manufactured infant formulas which contained lactose as the only source of carbohydrates, with 3/4 of the European-manufactured infant formulas being goat milk-based for unspecified reasons.

Complete weaning from formula and breast milk was achieved successfully by 1 year of age in 20/37 (54%) patients, while 17/37 (46%) continued to drink infant formula and/or breast milk beyond 1 year (range: 13 months to 35 months). The age of introduction of solids was between 4–6 months (9/37, 24%) and 6–8 months (27/37, 73%). One child who was breastfed until 35 months was delayed in accepting solid foods beyond 12 months.

### 3.2. Feeding/Swallowing Assessment

CFA was completed at the initial clinic encounter between 5 months and 2 years and 9 months (median age: 9 months) in 19 children. All 19 of these children were able to derive 100% of their nutrition orally. The majority of children (11/19, 58%) were determined to have normal feeding/swallowing, while 8/19 (42%) received a diagnosis of oropharyngeal dysphagia. In the eight patients with dysphagia, severity was mild in seven and mild–moderate in one. Clinical signs of aspiration such as coughing after drinking was observed in 7/8 (88%) of these children, though clinical concern for aspiration was relatively low as VFSS was never recommended. FOIS-P scores of 6 indicating total oral diet with no restriction relative to peers were present in 14 (74%). Four patients (21%) had FOIS-P scores of 5, indicating total oral diet without special preparation but with compensations required. The FOIS-P score was 4.5 in one participant (5%), indicating a total oral diet requiring special preparation of solids with or without compensations. Proxy-reported measures were available in eighteen including the NeoEAT-bottle in two and the PediEAT in sixteen; these scores were unremarkable in fifteen. Only the participant diagnosed with mild–moderate oropharyngeal dysphagia had an abnormal score which indicated a high level of caregiver concern regarding feeding/swallowing (PediEAT score: 151). Follow-up assessment of feeding/swallowing was recommended in all patients diagnosed with dysphagia. The initiation of feeding therapy was recommended in two patients and in the child with the most severe dysphagia, it was recommended that the frequency of local feeding therapy be increased.

### 3.3. Foods Encouraged and Discouraged at Different Age Groups

Parents’ approach to feeding children in terms of food encouraged and discouraged or avoided at different age groups was assessed. Data were available on feeding practices for 1 to <2 years on 37 children, between 2 and <4 years on 25 children, and between 4 and <6 years on 12 children. None of the parents in any age group reported complete elimination of any food group. The approach most commonly used was mostly limiting or not limiting the type and amount of carbohydrates and encouraging protein intake for a few children. No restriction of any type was reported in the diets of 10/37 children (27%) between 1 and <2 years, in 5/25 children (25%) between 2 and <4 years, and 3/12 children (25%) between 4 and <6 years. Foods with added sugar (candy, desserts, sugary drinks, fruit juice, etc.) and processed foods were discouraged in the diets of 17/37 children (46%) between 1 and <2 years, 15/25 children (60%) between 2 and <4 years, and 7/12 children (58%) between 4 and <6 years. The average fruit consumption was two servings for children 1 to <2 years of age, three servings for children 2 to <4 years of age, and consumption as high as 5–7 servings was noted in the 4 to <6-year groups. A few parents reported limiting starch in the diets of children 1 to <2 years of age (4/37; 11%) and 2 to <4 years of age (5/25, 20%), and none in the 4 to <6 years of age group. Protein-rich foods were encouraged in the diets of 2/37 children (5%) between 1 and <2 years of age; this number increased to 5/25 (20%) children between 2 and <4 years of age and 3/12 (25%) between 4 and <6 years of age. Parents’ perception of their children being on a special diet for Pompe disease was “not on special diet” in 25/37 children (68%) and “on a special diet” in 10/37 children (27%). The remaining parents (2/37, 5%) were not sure if their children were or were not on a special diet.

### 3.4. Basis of Food Selection

Parents used many resources to decide upon the dietary approach to feed their children with LOPD. Some parents (9/37, 24%) only relied on their own nutrition knowledge and beliefs regarding a suitable diet for their children, whereas the majority of parents (16/37, 43%) used one or more of the other resources, such as internet research, in addition to their own knowledge and beliefs. The remaining parents (12/37, 32%) reported using social media, support groups for Pompe disease, and support from family and friends to make their decisions on how to feed their children. Guidance from a registered dietitian and/or physician was used by a minority of parents (10/37, 27%).

### 3.5. Growth Percentiles and Dietary Intake

Weight of children between 1 and <2 years of age (*n* = 11) ranged between the 4th and 93rd percentile (median weight: 50th percentile) and height ranged between the 3rd and 97th percentile (median height: 61st percentile). Body mass index (BMI) of the children was between 3rd and 95th percentile (median BMI: 71st percentile). A total of 3/11 (27%) children fell in the overweight category (Table 1). Average calorie intake was 104 kcal/kg (range: 63–178 kcal/kg, median 97 kcal/kg; RDA for 12–35 months: 80–82 kcal/kg) and average protein intake was 5 g/kg (range: 2.5–7 g/kg; median 4.55 g/kg; RDA for 12–35 months: 1.5–1.1 g/kg). On average, carbohydrates provided 43% of daily total calories (range: 23–58%; AMDR for all ages: 45–65%), protein provided 19% of daily total calories (range: 14–27%; AMDR for 1–3 years: 5–20%), and fats provided 39% of daily total calories (range: 30–58%; AMDR for 1–3 years: 30–40%) (Figure 1A–C).

Weight of children 2 to <4 years of age (*n* = 14) ranged from 11th to 97th percentile (median weight: 66th percentile) and height ranged between 12th and 96th percentile (median height: 60th percentile). BMI ranged between 17th and 99th percentile (median BMI: 60th percentile). In total, 1/14 (7%) fell in the overweight category and 1/14 (7%) was in the obese category. Average calorie intake was 94 kcal/kg (range: 73–114 kcal/kg; median 92 kcal/kg; RDA for 13 months to <4 years: 82–85 kcal/kg) and average protein intake was 4.3 g/kg (range: 2.4–5.2 g/kg; median 4.19 g/kg; RDA for 13 months to <4 years: 0.95–1.1 g/kg). On an average, carbohydrates provided 44% of daily total calories (range: 22–62%; AMDR for all ages: 45–65%), while protein provided 18% of daily total calories (range: 13–25%; AMDR for 1–3 years: 5–20%), and fats provided 39% (range: 25–60%; AMDR for 1–3 years: 30–40%).

Lastly, the weight of children 4 to <6 years of age (*n* = 12) ranged from 14th percentile to 99th percentile (median weight: 63rd percentile) and height ranged between 24th percentile and 94th percentile (median height: 47th percentile). BMI was between 3rd and 98th percentile (median BMI: 53rd percentile). In total, 2/12 (17%) were in the overweight category and 2/12 (17%) were in the obese category. Average calorie intake was 92 kcal/kg (range 47–153 kcal/kg; median 90 kcal/kg; RDA for 4–7 years: 70–64 kcal/kg) and average protein intake was 3.9 g/kg (range: 1.6–6.6 g/kg; median 3.51 g/kg; RDA for 4–7 years: 0.95 g/kg). On an average, carbohydrates provided 49% of total daily calories (range: 37–63%; AMDR for all ages: 45–65%), protein provided 17% (range: 11–26%; AMDR for 4–18 years: 10–30%), and fats provided 34% of total daily calories (range 24–40%; AMDR for 4–18 years: 25–35%).

In addition, the head circumferences of all children between 1 and 3 years were within normal ranges (*n* = 17, between 9–98th), with a median head circumference of the 43rd percentile.

### 3.6. Laboratory Measurements

At the time of assessment, blood CK levels were available for twenty-six children, seven of which were on ERT (Figure 1D). CK levels ranged between 70 and 178 U/L (average: 129 U/L) in children 1 to <2 years of age (*n* = 6); between 47 and 676 U/L (average: 222 U/L) in children between 2 to <4 years (*n* = 11), and between 52 and 180 U/L (average: 118 U/L) in children 4 to <6 years (*n* = 9). Two children between 2 and <4 years had elevated CK levels, one of whom was started on ERT soon after this assessment. None of the children on ongoing ERT at the time of the study had high CK levels.

BUN: creatinine ratios were available for twenty-three children, four of which were on ERT (Figure 1E). BUN was increased in only two subjects (1 year and 2.2 years of age) with normal creatinine in all children. All four children between 1 and <2 years on whom data were available had an elevated BUN: creatinine ratio (range: 33–53). Out of the eleven children between 2 and <4 years on whom data were available, five had an elevated ratio (range: 28–78), and out of the eight children who were 4 to <6 years of age and for whom data were available, two had an elevated ratio (range: 32–54).

Urine Glc4 was collected at clinic visits and measured by Duke University Biochemical Genetics Laboratory. Data were available on twenty-eight children (1 to <2 years *n* = 9; 2 to <4 years *n* = 10; 4 to <6 years *n* = 9), eight of whom were on ERT (Figure 1F). Normal urine Glc4 (normal range 1–3 years: <8.3 mmol/mol creatinine, >3 years: <3.0 mmol/mol creatinine) was observed in all nine children between 1 and <2 years of age, 8/10 children between 2 to <4 years of age, and 5/9 children between 4 to <6 years of age. None of the six children with elevated urine Glc4 levels were on ERT.

When assessing relationships between macronutrient intake and lab values reflecting metabolic control, there were no statistically significant relationships (Appendix A).

## 4. Discussion

An acknowledgement of the role of diet therapy in the treatment of Pompe disease has prevailed as understanding of the pathogenesis of muscle damage in this lysosomal disease has increased. Historically, when muscle damage was mostly attributed to the massive accumulation of glycogen in the lysosomes, a high-protein and low-carbohydrate diet was implemented to reduce the glycogen burden and improve protein synthesis, to allow for an alternative fuel source and to prevent endogenous protein breakdown [21]. This yielded mixed results in a small number of adults with LOPD [22,23,24]. Along with a high-protein diet, supplementation with a branch chain amino acids was tried and showed improvement in respiratory function and muscle strength in an adult with LOPD [25]. In 2006, ERT for Pompe became available and we have continued to learn from both the success and limitations of this life-changing therapy, particularly regarding the functions of lysosomes and pathophysiology of the disease at the molecular level. Defective autophagy, dysregulation of lysosome-based signaling pathways, mitochondrial dysfunction, and oxidative stress are now considered as important in the pathophysiology of disease [4] and are also targets for improving treatment for Pompe disease. L-alanine supplementation along with ERT was shown to benefit a child with LOPD in improving body composition and resting energy expenditure [26]. A high-protein and low-carbohydrate diet with exercise therapy has shown benefit in terms of improved exercise tolerance, muscle enzymes, pulmonary function, and quality of life in adult LOPD patients on ERT and continues to be recommended as adjunct therapy for adults with LOPD [5]. Use of exogenous ketone precursors and various antioxidants as adjunct therapies to ERT was shown to enhance autophagic clearance and improve muscle pathology in GAA-KO mice [27]. The safety and efficacy of the high-protein, low-carbohydrate diet in children with Pompe disease has not been comprehensively studied, and guidance is particularly needed for the dietary management of very young children with LOPD who are diagnosed via NBS. Given that there are no formal guidelines on when to initiate ERT in this population, parents have turned to dietary adjustments in hopes of delaying the onset and/or severity of disease. This study evaluated the feeding practices, dietary intake, growth, metabolic control, and feeding/swallowing in a cohort of young children diagnosed with LOPD through NBS with the aim of supporting future nutrition guidelines for children with Pompe disease.

In this study, breastfeeding was initiated in 73% of infants with LOPD. Although this is lower than the 2019 national breastfeeding initiation rate of ~83% [28], it supports that infants with LOPD can be successfully breastfed. Also, once initiated, breastfeeding alone or in addition to infant formula was continued until age 1 year in 78% of infants with LOPD, a higher number than the national average of ~36% [28]. Infants with LOPD who received formula were primarily given standard infant formulas containing lactose, maltodextrin, and corn syrup solids. However, some infant formulas contained sucrose making up ~20% of the total carbohydrate content, and ~10% of infants with LOPD received European formulas which are free of sucrose, glucose, and genetically modified organisms (GMOs). Avoiding infant formulas with sucrose in Pompe disease may be considered good practice considering the pathophysiology of the disease, although this is theoretical and prospective studies are warranted.

Oropharyngeal dysphagia is increasingly recognized as an important feature of LOPD in adults [29,30,31,32,33]. Swallowing difficulty is attributed to bulbar muscle involvement and lingual muscle weakness, for example, is known to occur commonly, is often present early in the disease course, and may be severe in some adults with LOPD [29,30,31,34]. Clinical assessment of feeding/swallowing in a subset of our cohort suggests that dysphagia is relatively common in children with LOPD diagnosed via NBS. In the 19 patients who received CFA, 42% received a diagnosis of oropharyngeal dysphagia and clinical signs of aspiration were present in 37%. However, the severity of dysphagia was typically mild, the presence of dysphagia rarely affected the ability to take an unmodified oral diet, and despite the observation of clinical signs of aspiration, overall concern for aspiration was relatively low and VFSS was never recommended. Despite this, routine assessment of feeding/swallowing in children with LOPD diagnosed via NBS should be conducted for several reasons. One, diagnosis of oropharyngeal dysphagia is an important sign of disease in terms of monitoring the need for treatment. Two, despite the mild severity of oropharyngeal dysphagia we typically encountered, the diagnosis of dysphagia resulted in feeding recommendations and caregiver education. Three, in patients with dysphagia, follow-up feeding/swallowing assessment was recommended, including the possible need for VFSS if clinical signs of aspiration persisted. Four, important recommendations derived from the CFA in this population included the recommendation of feeding therapy in two patients and an increased frequency of feeding therapy in one patient including recommended goals for local providers.

Regarding the introduction of solids, all children except for one child were weaned from breast milk and formula and introduced to solid foods successfully and when developmentally appropriate. After the introduction of solid foods, the majority of parents (75%) of children with LOPD used their own knowledge and understanding of nutrition when deciding what to feed their children while also seeking support from internet research, other Pompe families and support groups, and family and friends. In fact, less than 30% of parents reported consulting a dietitian or physician for nutritional guidance. This likely reflects an underutilization of registered dietitian services during their clinical appointments and/or the lack of recognition of the role nutrition plays in managing Pompe disease.

The goals of nutrition therapy in Pompe disease in children are to meet their nutritional needs, promote optimum growth and development, and maximize muscle health while reducing the glycogen burden in the body. Careful attention needs to be given to the composition of the diet to meet these nutritional goals. All food groups have nutritional value and hence can contribute to a healthy diet for children with Pompe disease. Limiting carbohydrates and particularly avoiding simple carbohydrates is considered helpful, potentially to limit glycogen accumulation. It was reassuring to know that the diets of children with LOPD in this study contained all food groups and that no food group was completely eliminated in their diets. Although ~25% parents reported not having any dietary restrictions, ~/>50% reported limiting simple sugars, confectionaries, fruit juice, and processed foods. Implementing this strategy in early childhood and continuing with this approach later is helpful to meet the goal of overall carbohydrate reduction in Pompe disease as well as prevent obesity. The benefit of limiting or avoiding fruits in the Pompe diet is often questioned in clinic by parents (by authors’ personal experience) due to the fructose content. Although some children consumed an ad lib amount of fruit daily, most children in the study consumed fruits in moderation (2–3 servings/day) and this is important for their nutritional value as well as variety in the diet, especially when sugary foods are being restricted. Avoiding fruit juice and spacing fruit consumption may be helpful to avoid a fructose load, although it is not known if this affects glycogen synthesis or accumulation rates. Vegetable consumption was noted in the 24 h food records of 29/37 (78%) children in the study and ranged from 1–4 servings/day. Our data is more encouraging than the average vegetable, fruit, and sugar-sweetened beverage consumption in children 1–5 years in the national survey [35].

Our data shows that children with LOPD diagnosed by NBS have normal growth between 1 and 5 years and are meeting their calorie needs via oral nutrition. The overall incidence of being overweight and obese in children with LOPD in our study was slightly lower (25%) than observed in the general population (1/3 or ~33%) [36], potentially because of vigilance of parents in restricting sugar-containing and processed foods. However, there was an increase in the number of LOPD children being classified in the obese category based on BMI as they got older: 0% in the 1-to-2-year age group, 7% in the 2-to-<4-year age group, to 17% in the 4-to-<6-year age group. Avoiding obesity is important, particularly in neuromuscular disorders such as Pompe disease as it affects mobility, posture, balance and adds to the risk of falls. It is also unclear how obesity and its associated metabolic shifts could alter and potentially worsen glycogen storage and metabolic derangements in Pompe disease, which further supports the importance of maintaining a healthy BMI starting at a younger age.

Regarding diet composition, there was wide variation in the percentage of calories from carbohydrates within each age group. The average calorie intake from carbohydrates (43–44%) was below the lower end of the range recommended in the Dietary Guidelines for Americans (45–65%) [37] until 4 years, likely because of restriction in simple sugar intake in the diets of these children. Average fat intake was within recommended ranges in all age groups. Protein intake varied widely within each age group and although the average percentage of calories from protein was within the range recommended [37], protein intake was 3–4 times higher than the RDA across all age groups. Our data are consistent with other surveys that show that in the US [38] and other developed countries [39], children’s and adolescent’s protein intake trends are usually 2- to 3-fold higher than the recommendations. Yet, higher protein intake in infancy is associated with obesity risks later in life [39,40,41]. There is a scarcity of data regarding what the optimum protein intake in children is, the effect of high protein intake on lean body mass versus fat mass, the effect of animal versus plant protein, the effect of age and gender [42]. Some studies suggest a positive trend between high protein intake and BMI on account of the increase in fat-free mass in mid-childhood which has positive effects on growth and bone mass [43]. Indeed, the positive impact of increased protein intake early in life would certainly be beneficial in children with LOPD who are more likely to experience progressive muscle weakness as they grow, resulting in reduced motor abilities.

Among the lab data collected, blood CK and urine Glc4 are relevant to the severity of disease and efficacy of ERT. LOPD children on ERT had normal CK and children with elevated CK and urine Glc4 were not on ERT at the time of this survey. For children who were on ERT, CK and urine Glc4 data before beginning ERT was not available for this study. BUN, creatinine, and their ratio are markers of hydration and renal function, which can be affected by protein intake. Children in the present study consumed 3–4 times the RDA of protein and 11/23 (48%) had elevated BUN/creatinine ratio. All four children between 1 and <2 years of age whose BUN/creatinine ratio was available had an elevated ratio. Based on this observation and association of increased protein intake in infancy to obesity later in life, we recommend that a cautious approach be taken regarding protein content of children with LOPD <2 years of age. Protein tolerance as reflected by BUN and creatinine ratio appeared to improve with age in the study cohort. This highlights the importance of individualizing and optimizing protein intake in children with LOPD based on their age and protein tolerance.

Based on the nutrition goals for children with Pompe disease mentioned previously as well as the findings in this study, we provide the following framework for the recommended nutritional guidelines for children <6 years of age with LOPD:Breastfeeding, because of its superior nutritional composition, immune-protective properties, and positive impact on growth and neurocognitive development, should be allowed in infants with LOPD based on the ability and preference of the mother and child.Standard infant formulas are suitable for meeting the nutritional needs of children with LOPD for proper growth. The benefit of avoiding sucrose and simple sugars in infant formulas in LOPD is not known at this time but these should be avoided if an alternate and affordable formula without sucrose and simple sugars is available.Clinical assessment of feeding/swallowing should be completed in children with LOPD diagnosed via NBS to determine the presence or absence of dysphagia, as well as the need for VFSS, feeding therapy, and follow-up assessment.Diets of children with LOPD should include all food groups and should be rich in whole grains, vegetables, dairy, meats and fruits in age-appropriate serving sizes.Foods with high sugar content as well as refined and highly processed foods should be avoided to prevent excessive carbohydrate intake and to prevent obesity.Assessment of growth, BMI, and nutritional intake should be part of ongoing management of young children with LOPD.Protein intake in children with LOPD should be individualized and based on tolerance. A gradual transition towards a protein-rich diet should start in childhood while monitoring tolerance.Periodic consultation with a registered dietitian with expertise in inherited metabolic disorders is recommended for nutrition education and appropriate diet prescription.CK, comprehensive metabolic panel (CMP), urine Glc4 should be monitored as part of ongoing disease management.

Limitations of the study: It is important to acknowledge that given the observational, cross-sectional nature of this study, there were inherent limitations. First, diet analysis was based on 24 h recall which has the limitations of memory of the recaller, challenges in accurately estimating portion sizes, potential of misreporting and not being reflective of an individual’s typical diet. For future studies, three-day diet log maintained over two weekdays and one weekend day would be more appropriate to use as it offers a more reliable snapshot of an individual’s habitual eating and is not memory dependent. The variability in socio-economic status, education level, access to food, medical resources, and social and family support of parents could have contributed to the differences in feeding patterns and dietary intake of children. Additional studies of a similar framework are warranted to determine how our findings compare to those from other centers. We also want to acknowledge that while the FOIS-P and the proxy measurements NeoEAT breastfeeding, NeoEat bottle feeding, and PediEAT are widely used in clinical setting with many disease populations, their use has not been specifically validated for use in Pompe disease and other inherited metabolic disorders. Additionally, only 28% of our study population was on ERT at the time of assessments, so it was not possible for us to compare results between those who were receiving ERT to those who were not. Future prospective and longitudinal studies should investigate the impact of ERT on parent food choices and the role of nutrition therapy as an adjunct to ERT.

## 5. Conclusions

Children with LOPD diagnosed via NBS have normal growth and while a majority have normal swallowing and feeding ability, some have mild dysphagia. They are consuming a diet that is somewhat restricted in carbohydrates and their protein intake is 3-4 times the RDA for age. No difference is noted in the growth, macronutrient intake and lab parameters in this small cohort based on their ERT status. Results of this study show a need for performing an individualized assessment of the dietary intake and prescribing a diet based on their feeding ability, growth and laboratory parameters. This study provides a framework for developing future nutrition guidelines for children with LOPD. 

## Figures and Tables

**Figure 1 nutrients-17-01909-f001:**
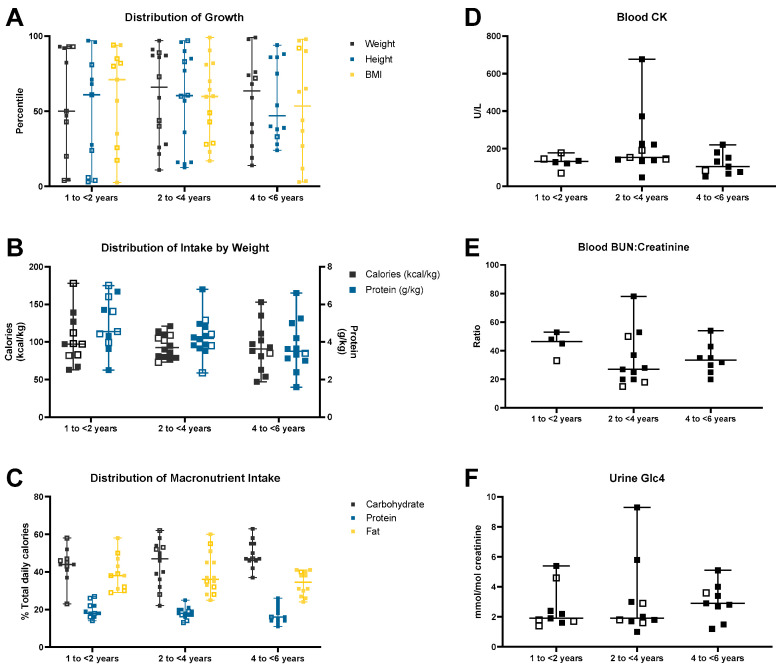
Growth, macronutrient intake, and clinical chemistry reports of children with LOPD identified through newborn screening. The distribution of data for each age group are depicted. The bars indicate the median and range. Data points from individuals receiving enzyme replacement therapy (open square) are separated from those not receiving enzyme replacement therapy (closed square).

**Table 1 nutrients-17-01909-t001:** Growth and dietary intake of children with LOPD.

Age (years)	Weight (kg)	Weight (%tile)	Height (cm)	Height (%tile)	BMI	BMI (%tile)	BMI 85–<95%	BMI ≥ 95%	Calories/kg	Protein g/kg
1–<2, *n* = 11							3	-	RDA 80–82	RDA 1–1.5
Median	10.5	50	79	61	17	71			97	4.55
Range	7.9–13.5	4–93	71.5–86	3–97	14.2–18.9	3–95			63–178	2.5–7
2–<4, *n* = 14							1	1	RDA 70–85	RDA 0.95–1.1
Median	14.4	66	98.2	60	16.1	60			92	4.19
Range	11.4–20.5	11–97	84–110.1	12–96	14.7–17.6	17–99			73–114	2.4–5.2
4–<6, *n* = 12							2	2	RDA 70	RDA 0.95
Median	17.5	63	108.3	47	15.6	53			90	3.51
Range	14.8–26.2	14–99	100.5–115.4	24–94	14–20.1	3–98			47–153	1.6–6.6

## Data Availability

The data presented in this study are available on request from the corresponding author.

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
