# Peer review of "An Assessment of Dietary Intake, Feeding Practices, Growth, and Swallowing Function in Young Children with Late-Onset Pompe Disease: A Framework for Developing Nutrition Guidelines"

_nutrients, 2025, doi:10.3390/nu17111909_

Round 1
Reviewer 1 Report
Comments and Suggestions for Authors
Attached.

Author Response
|
Response 1: Thank you for pointing this out. We have corrected the word “late-onset” in the title of the paper.
|
|
Comments 2: CMP is mentioned for the first time on page 10. Authors should give all the abbreviations in one place. What are they? Same with RDA mentioned in the abstract section. When the acronym is noted for the first time, the full name should be written.
|
|
Response 2: We thank you for bringing this to our attention and agree with your suggestion. We have listed all the abbreviations used in the paper on page 10, starting on line 446 per the journals protocol. We have provided the full name of the abbreviations of all acronyms the first time it was used starting with the abstract and throughout the manuscript.
Comments 3: I believe Urine Hex4 is the same as urine Glc4. Response 3: We agree urine Hex4 and Glc4 are the same and apologize for this confusion. We have noted that instead of Glc4 we had used Hex4 in the “framework for guidelines” table on page 10. We have now consistently used Glc4 across the whole manuscript.
Comments 4: Figure 1 is not cited in the text except for Figure 1D. The same applies to Table 1, which is not being cited in the text. Response 4: We apologize for this error and thank the reviewers for pointing this out. We have now mentioned Figure 1 (A-F) in the materials and methods section on page 2, line 103 of the manuscript. We have now also mentioned Table 1 in the materials and methods section on page 2, line 96.
Comments 5: Protein intake was 3-4 times higher than the RDA. Where is this number coming from? The RDA for 4-7 years is 0.95 g/kg. Is this standard? Response 5: We are sorry about this confusion and thank the reviewer for the opportunity to elaborate on this point. The diets of 37 children with LOPD were analyzed based on their 24 hr food recall. This diet analysis showed the protein intake of these children to be 3-4 times higher than the RDA. The range of and median protein intake for each age group is mentioned in the “results” section 3.5 under “growth percentiles and dietary intake” on page 4 (paragraph 4) and 5 (paragraph 1 and 2). Comparison of the protein intake with RDA for protein for each age group is also shown in Table 1 on page 6. The RDA for protein for 4-7 years is 0.95/kg as a standard based on National Academy of Sciences, Institute of Medicine, Food and Nutrition Board, 2006.
Comments 6: It is unclear how these 37 children were classified into LOPD in the method section. Response 6: Thank you for bringing this very important point to our attention. We agree more information is needed regarding this in the article. We have now provided the information on the classification of these children as LOPD in the “materials and methods” section on page 2, paragraph 3 lines 66-70.
Comments 7: There is no mention of the effect of exercise on LOPD if any.
Response 7: The authors acknowledge the role of exercise therapy in Pompe disease and would kindly like to draw attention to the “introduction” section on page 2, paragraph 2 and lines 55-56 as well as “discussion” section on page 7, paragraph 1, lines 266-269. References on this topic are also listed in the bibliography. We agree that exercise intervention was not tried in this cohort of LOPD patients due to their young age.
|

Reviewer 2 Report
Comments and Suggestions for Authors
-
Add a brief paragraph acknowledging the limitations of using a single 24-hour dietary recall and recommend improved methodologies for future studies.
-
Explore whether there are any notable dietary practice differences between children receiving ERT and those not.
-
Include brief validation context for the tools used (NeoEAT, FOIS-P) in populations with metabolic disorders, if available.
Author Response
|
Comment 1: Add a brief paragraph acknowledging the limitations of using a single 24-hour dietary recall and recommend improved methodologies for future studies. |
|
Response 1: Thank you for pointing this out. We agree on elaborating on the limitations of the 24 hr dietary recall. Therefore, we have acknowledged this limitation on page 10, paragraph 1 and in lines 397-402 of the article while also recommending the use of 3 day diet recall in future studies.
|
|
Comment 2: Explore whether there are any notable dietary practice differences between children receiving ERT and those not. |
|
Response 2: Thank you for your comment. We agree that it is important to consider the differences in nutrient intake in those on ERT compared to those not on ERT. We kindly refer the reviewer to Figure 1C which shows the distribution of macronutrient intake between the ERT and non-ERT cohorts. Based on our findings, there were no observable differences in carbohydrate, protein, or fat intake between those on ERT versus not on ERT. However, we do emphasize in the limitations section that our cohort on ERT was small, and therefore we cannot draw further conclusions.
|
|
Comments 3: Include brief validation context for the tools used (NeoEAT, FOIS-P) in populations with metabolic disorders, if available. |
|
Response 3: We agree with the reviewers regarding the importance of the validity of these tools specifically in metabolic disorders. Although widely used in clinical practice in a variety of settings and with many disease populations, the NeoEAT, FOIS-P tools have not been specifically validated for use in Pompe disease and other inherited metabolic disorders. We have acknowledged as a limitation to our study on page 10, paragraph 1 and in lines 406-410.
|

Reviewer 3 Report
Comments and Suggestions for Authors
The work presented by Pendyal et al. is very well written and meticulously organized. The authors provide a comprehensive data presentation, covering all major aspects of the study in deep detail. The same is true for the discussion section in the manuscript, where the authors reflect to every single detail of the presented data in order to define the excellent framework for recommended nutritional guidelines at the end of the discussion.
Comment:
- As the authors point out, various studies need to be performed in individuals with this disease to derive important information for helping parents to apply the most appropriate diet to their children and improve life quality and expectancy. From a metabolic standpoint I could envisage a need for the follow up of the process of gluconeogenesis in situations of increased protein/aminoacid intake. Most of the aminoacids are either glucogenic or partially glucogenic, with the exception of lysine and leucine, which are purely ketogenic. Thus, the substitution of carbohydrate by protein and the age at which that needs to occur requires appropriate evaluation of this gluconeogenic pathway. The same would be true for the glycogenesis pathway, to determine how impactful the excess protein intake would be on glycogen stores. With a prevalence of 1 in 20,000 this disease affects many individuals worldwide that would certainly benefit from dietetic guidelines supported by metabolic research.
Minor comments:
- Page 3, line 135: “14 74%)” should be 14 (74%);
- Page 7, ine 288: “muascle”; correct to muscle;
- Page 10, line 397: “between those were received ERT”; this segment in the sentence needs correction.
Author Response
|
Response 1: Thank you for pointing this mistake out. We have made this correction in the FOIS-P score of children to 14 (74%). We want to kindly notify you that this information now appears on page 3, paragraph 4 and in line 143 [vs 135] due to some edits made in the previous section.
|
|
Comments 2: Page 7, line 288: “muascle”; correct to muscle |
|
Response 2: We apologize for this spelling error. We have, corrected this error by correctly spelling the word “muscle”. The correction now appears on page 7, paragraph 3, line 294 of the revised manuscript.
|
|
Comments 3: Page 10, line 397: “between those were received ERT”; this segment in the sentence needs correction. |
|
Response 3: Thank you pointing out this mistake in the sentence structure. This sentence which acknowledges the limitations of the study to not being to compare results between children who were on ERT and those who were not on ERT, now appears on page 10, paragraph 1 and line2 410-412 of the revised manuscript.
|
